# Mutual influence between language and perception in multi-agent communication games

Xenia Ohmer[1]*, Michael Marino[1], Michael Franke[1,2], Peter König[1,3]

**1** Institute of Cognitive Science, University of Osnabrück, Osnabrück, Germany, **2** Department of Linguistics, University of Tübingen, Tübingen, Germany, **3** Department of Neurophysiology and Pathophysiology, University Medical Center Hamburg-Eppendorf, Hamburg, Germany

☉ These authors contributed equally to this work.
* xenia.ohmer@uni-osnabrueck.de

**Data Availability Statement:** All code and generated data are available at the Open Science Framework (OSF): https://osf.io/qu4xp/.

**Funding:** XO and MM acknowledge funding from the Deutsche Forschungsgemeinschaft (DFG,

## Abstract

Language interfaces with many other cognitive domains. This paper explores how interactions at these interfaces can be studied with deep learning methods, focusing on the relation between language emergence and visual perception. To model the emergence of language, a sender and a receiver agent are trained on a reference game. The agents are implemented as deep neural networks, with dedicated vision and language modules. Motivated by the mutual influence between language and perception in cognition, we apply systematic manipulations to the agents' (i) visual representations, to analyze the effects on emergent communication, and (ii) communication protocols, to analyze the effects on visual representations. Our analyses show that perceptual biases shape semantic categorization and communicative content. Conversely, if the communication protocol partitions object space along certain attributes, agents learn to represent visual information about these attributes more accurately, and the representations of communication partners align. Finally, an evolutionary analysis suggests that visual representations may be shaped in part to facilitate the communication of environmentally relevant distinctions. Aside from accounting for co-adaptation effects between language and perception, our results point out ways to modulate and improve visual representation learning and emergent communication in artificial agents.

## Author summary

Language is grounded in the world and used to coordinate and achieve common objectives. We simulate grounded, interactive language use with a communication game. A sender refers to an object in the environment and if the receiver selects the correct object both agents are rewarded. By practicing the game, the agents develop their own communication protocol. We use this setup to study interactions between emerging language and visual perception. Agents are implemented as neural networks with dedicated vision modules to process images of objects. By manipulating their visual representations we can show how variations in perception are reflected in linguistic variations. Conversely, we

German Research Foundation) - GRK 2340. The funders had no role in study design, data collection and analysis, decision to publish, or preparation of the manuscript.

**Competing interests:** The authors have declared that no competing interests exist.

demonstrate that differences in language are reflected in the agents' visual representations. Our simulations mirror several empirically observed phenomena: labels for concrete objects and properties (e.g., "striped", "bowl") group together visually similar objects, object representations adapt to the categories imposed by language, and representational spaces between communication partners align. In addition, an evolutionary analysis suggests that visual representations may be shaped, in part, to facilitate communication about environmentally relevant information. In sum, we use communication games with neural network agents to model co-adaptation effects between language and visual perception. Future work could apply this computational framework to other interfaces between language and cognition.

## Introduction

Language is not an isolated system. Language is grounded in the physical world and serves to coordinate and achieve common objectives [1, 2]. Under this functional perspective, it becomes obvious that language interfaces with many areas of cognition, among others, perception, action and embodiment, and social cognition [3]. To understand the origins and evolution of language it is important to take these connections into account. In this paper, we demonstrate how deep learning models of interactive language emergence can be used to study the relationship between language and other areas of cognition, focusing on the interface between language and visual perception.

Deep neural networks (DNNs), even though originally developed for engineering purposes, have been used to study human cognition in various fields. In terms of language emergence and language evolution, simulations with neural network agents have been used to model, for example, the emergence of color naming systems [4, 5], contact linguistic phenomena [6], the emergence of word learning biases [7, 8], or the emergence of compositional structure [9–11]. In terms of visual perception and representation learning, DNNs have been used to model brain activations in the visual cortex [12–14] and judgments of image similarity [15, 16]. Our work extends existing research by studying *interactions* between language emergence and visual representation learning in neural network agents.

In human cognition, the influence between language and perception is bidirectional. Expressions for concrete concepts like colors depend on perception [17]. But also abstract concepts can be understood and represented via metaphoric mappings to concrete concepts grounded in sensorimotor experience, for example in reasoning about time as a moving object ("The time will come when . . .", "Time flies") [18]. Similarly, the effects of language on perception can be observed for high-level cognitive processes such as recognition as well as low-level processes such as discrimination and detection [19]. In particular, language affects perceptual processing by imposing categorical structure [20, 21]. We aim to analyze such bidirectional influences systematically, by studying the effects of variations in visual representations on emergent communication and vice versa.

More precisely, this paper looks at three questions: (i) how does perceptual bias affect language emergence, (ii) how does exposure to a particular linguistic input influence perceptual representations, and relatedly (iii) could perceptual representations be shaped by an optimization process towards successful communication of environmentally relevant distinctions. We use a conventional language emergence setup with two agents, a sender and a receiver, playing a reference game, based on the signaling game originally developed by Lewis [1]. The sender sees a target object and sends a message to the receiver. Using that message, the receiver tries

to identify the target among a set of distractor objects. By choosing this kind of game, we study the emergence and effects of *referential labels*, with sets of real-world objects as denotations. Reference is arguably a core function of language around which more complex functions are organized [22]. The agents have a vision module to process input images, and a language module to generate (sender) or interpret (receiver) messages. In line with many existing models [23–25], the vision modules are implemented as pretrained convolutional neural networks (CNNs) and the language modules as recurrent neural networks (RNNs). The following three paragraphs enlarge on how this setup is adjusted to address each question.

(i) To study the influence of perception on language, we design agents with different visual biases, such that object representations vary between agents. We fix these biases and combine different agents to quantify differences in the emergent communication protocols. Given that concept formation in humans depends on perceptual similarity [26], our manipulations target the similarity relationships between object representations. By applying a new method called *relational label smoothing* to the CNN pre-training we modify the class labels, such that the resulting representational similarities between objects vary for different conditions. Thereby, we can test how language grounding is influenced by these differences, and how certain perceptual predispositions can benefit communication.

(ii) To study the influence of language on perception, we allow agents to adapt their visual representations (CNN weights) while playing the communication game. We measure how perception adapts to fixed languages in language learning, or to different communication partners in language emergence. To analyze changes in perception we again rely on similarity relationships between visual representations. Several studies concerning categorical perception have shown that language affects perceptual similarity [19]. Moreover, developing a system of similarity relationships along *relevant* perceptual dimensions (e.g., color, shape, magnitude, texture) is a major achievement in child development [27]. In our case, relevance is determined by the communication game. Thus, our setup not only allows us to study how language influences perceptual similarity but also how a system of similarity relationships with respect to task-relevant dimensions can evolve via communication.

(iii) Finally, an evolutionary analysis explores whether an agent's perceptual system might be optimized over time to facilitate communication about relevant aspects of the environment. As in (i), we consider agents with different, fixed perceptual biases. We train an extensive variety of agent combinations on the reference game and derive a payoff matrix for a symmetric population game. We subject this payoff matrix to a simple analysis in terms of evolutionary stable states (ESSs) [28]. Thereby, we can determine whether certain perceptual representations (biases) are more likely to prevail in an adaptation process to the demands of linguistic interaction, which in our case defines the agents' environment. Importantly, ESS-analysis does not entail a commitment to an underlying process of biological evolution. ESSs can also be considered the rest points of other (agent-internal) optimization processes.

## Related work

Communication games have been used to study the emergence and evolution of language theoretically [29], experimentally [30, 31], and computationally [32]. Artificial intelligence research has also emphasized the importance of learning to communicate through interaction for developing agents that can coordinate with other, possibly human agents in a goal-directed and intelligent way [33]. It has been shown that by playing communication games, artificial (robotic) agents can self-organize symbolic systems that are grounded in sensorimotor interactions with the world and other agents [34–37]. For example, in a case study with color stimuli, simulated agents established color categories and labels by playing a (perceptual)

discrimination game, paired with a color reference game [36]. Bleys et al. extended these findings to robotic agents, demonstrating that successful color naming systems emerge despite differences in the agents' perspective [37]. These studies are mainly interested in how a categorical repertoire can become sufficiently shared among the members of a population to allow for successful communication. Our analyses, in contrast, assume that successful communication will emerge, and focus on how visual representations and language shape each other.

Over the past years, research using communication games to study language emergence in DNN agents has been gaining popularity [38]. Some of these models skip any form of perceptual processing by using symbolic input data [39–41]. Even though other models implement a visual processing system and work with image data [23, 42], they have rarely been used to explore the relation between language and visual perception. Notably, Rodriguez et al. examined the effects of natural differences in object appearance (such as frequency, position, and luminosity) on emergent communication [24]. Apart from that, Bouchacourt and Baroni measured the alignment between agents' internal representations and conceptual input properties to determine whether emergent language captures such properties or relies on low-level pixel information [43]. Still, these models usually extract object representations from fixed, pretrained CNNs. As a result, they make claims about how the emergent language relates to the input, not the visual perception of that input. In our work, we exploit the flexibility of modern setups and introduce systematic variations in the agents' visual processing, such that we can establish a relationship between differences in visual processing and differences in emergent protocols.

## Materials and methods

### Data set

We use the *3dshapes* data set [44]. The data set contains images of 3D shapes in an abstract room, generated from six latent factors, which can vary independently: floor color (10 values), wall color (10 values), object color (10 values), object scale (8 values), object shape (4 values), and object orientation (15 values). We use a subset of four different object colors (red, yellow, turquoise, purple), and four different object scales (equally spaced from smallest to largest); amounting to 96000 different images. For our purpose, we define objects by color, scale, and shape of the geometric shape, such that there are $4^3 = 64$ different objects. The term "object" refers to an object class, such as "tiny red cube", with each image representing an instance of such an object. Consequently, if we say that two agents see the same object, e.g., a tiny red cube, they both see an object that agrees on the relevant attributes (object color, object scale, and object shape), but not necessarily on the irrelevant ones (floor color, wall color, object orientation), e.g., they might both see a tiny red cube, one against a yellow wall and another against a green wall. Similarly, when we say that two objects are different, they differ in at least one of the relevant attributes but may agree on all irrelevant ones.

### Communication game

Two agents, sender *S* and receiver *R*, play a reference game where one round of the game proceeds as follows:

1. A random object is selected as the target.

2. *S* sees an image of the target and produces a message. Messages have length *L* and consist of a sequence of symbols $(s_1, \ldots, s_L)$ from vocabulary $V = \{0, \ldots, |V| - 1\}$.

3. $R$ sees a possibly different image of the target and additionally $k$ random distractor images, showing other objects. Based on the message from $S$, $R$ tries to select the image showing the target.

4. If $R$ succeeds, both agents receive a positive reward, $r = 1$, otherwise they receive zero reward, $r = 0$.

Three attributes—color, size, and shape—define what we call "object". Sender and receiver see potentially different images of the same target object, while the distractor images show different objects. Consequently, it lies in the nature of this game, that *conceptually relevant* (i.e. class-defining) attributes and *task-relevant* attributes coincide.

## Model

The model components and their interactions in the communication game are shown in [Fig 1]. Sender and receiver each have a vision module to process images, $i$, and a language module to generate (sender) or process (receiver) discrete messages, $m$. The sender maps the input image to a probability distribution over messages, $\pi_S(m \mid i)$, by sequentially generating a probability distribution across symbols conditioned on the symbols produced so far. The receiver maps the input message onto a probability distribution over (target and distractor) images, $\pi_R(i \mid m)$. These distributions define the agents' policies. During training, actions are sampled from the policies, whereas for testing the arguments of the maxima are used.

The vision module, $v(\cdot)$, is a CNN pretrained to classify the 64 different objects. The agents use the activations of the fully connected layer before the final softmax layer as object representation. The language module, $l(\cdot)$, consists of an embedding layer and a gated recurrent unit (GRU) layer [45]. Each agent has an additional fully connected layer, $f^1(\cdot)$, mapping the visual representations onto the same dimensionality as the GRU hidden state. For the sender, the output of $f_S^1$ is used to initialize the hidden state of the language module. The sender has an additional fully connected layer, $f_S^2(\cdot)$, mapping the GRU hidden state onto a probability

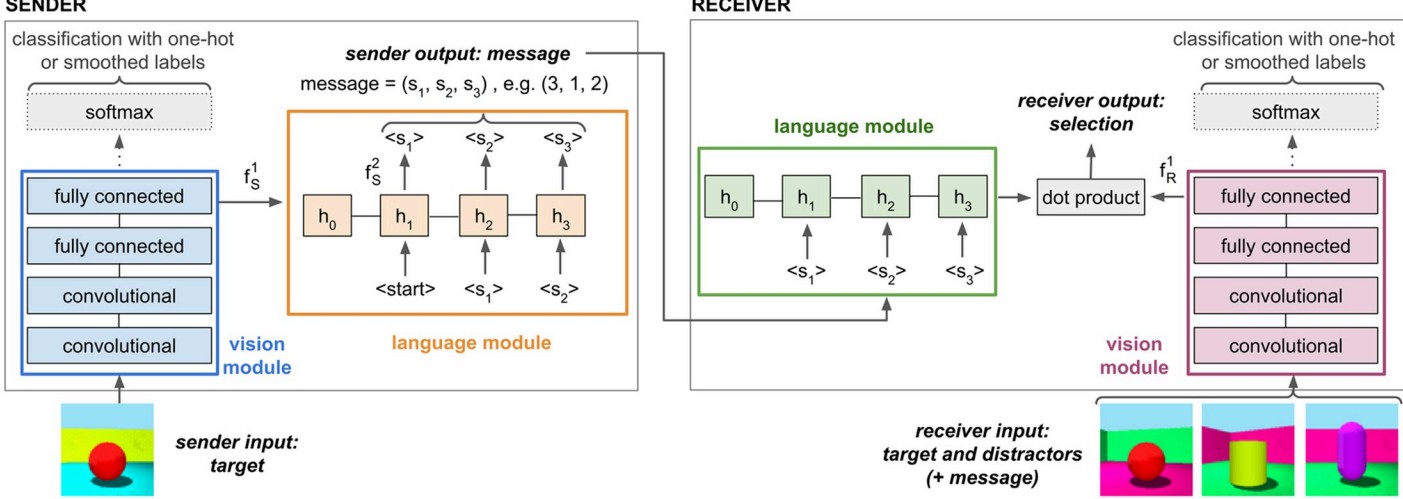

**Fig 1. Schematic visualization of sender and receiver architecture and their interaction in one round of the reference game.** The sender takes an image of the target object as input. The image is processed by the sender's vision module and the resulting activations are used to initialize the hidden state, $h_0$, of the sender's language module. The initial input to the sender's language module, ⟨start⟩, is a zero vector of the same dimensionality as the symbol embeddings, and at each time step a symbol is sampled from its output distribution. The generated message is processed by the receiver's language module. In addition, the target and the distractor images are processed by the receiver's vision module. The final selection probability is proportional to the dot product between the receiver's final hidden state and the image embeddings.

distribution across symbols at each time step, $t$, such that $\pi_S(m = (s_1, \ldots, s_L) \mid i) = \prod_{t=1}^{L}$ $\pi_S(s_t \mid s_{k<t}, i)$, with $\pi_S(s_t \mid s_{k<t}, i) \propto f_S^2(h_t)$. For the receiver, the dot product between the output of layer $f_R^1$ and the final GRU hidden state defines the selection policy: $\pi_R(i \mid m) \propto \exp$ $(f_R^1(v_R(i)) \cdot l_R(m))$.

## Introducing perceptual biases via relational label smoothing

In order to investigate the influence of differences in perception on emergent language, we develop a method called *relational label smoothing*, which allows us to systematically manipulate the CNN pretraining and thereby to create agents with different perceptual biases. We aim to have four conditions, next to the unmanipulated DEFAULT. Specific biases for either of the object-defining attributes—color, scale, and shape—make up three of these conditions. E.g., in the COLOR condition, color similarities are amplified. In addition, we experiment with an ALL condition, where we amplify similarities for all three attributes simultaneously.

Relational label smoothing calculates the target at training time as a weighted sum of the usual one-hot target, $\mathbf{y}_0$, and a relational component, $\mathbf{y}_r$, according to

$$\mathbf{y} = \sigma \mathbf{y}_r + (1 - \sigma)\mathbf{y}_0 \,,$$

where $\sigma \in \mathbb{R}$ is the smoothing factor, controlling the strength with which the relationship(s) should be enforced.

To enforce object similarities along one specific attribute (or dimension), $a$, we use a single-level hierarchical version of relational label smoothing. If $i$ is the true object class, we define superclass $C_i$ as the set of object classes having the same value as $i$ for $a$. Then $\mathbf{y}_r$ is given by

$$y_{r_{ij}} = \begin{cases} (n - 1)^{-1} & j \in C_i \text{ and } i \neq j \\ 0 & \text{else} \end{cases},$$

where $n$ is the number of object classes in $C_i$. E.g., in the COLOR condition, if the training sample is a red object, the relational component, $\mathbf{y}_r$, is a uniform distribution of $1/(16 - 1)$ across the class indices of the other 15 red objects, see Fig 2A, which increases the representational similarity between red objects, and analogously that of objects sharing other color values, see Fig 2B.

In order to enforce relationships for multiple attributes in a single model, we generalize the previous definition to include $\mathbf{y}_r$ to be a sum over relational components,

$$\mathbf{y}_r = \frac{1}{N} \sum_{a=1}^{N} \mathbf{y}_{r_a} \,,$$

where $N$ is the number of attribute relationships, and $\mathbf{y}_{r_a}$ represents the relational component from attribute $a$. To calculate the relational component for the ALL condition, we average the relational components from the COLOR, SCALE, and SHAPE conditions.

## Training and hyperparameters

We use a train/test split of 0.75/0.25.

**General setup.** The general training setup varies depending on which direction of influence between perception and language is being investigated. A schematic overview of these variations is shown in Fig 3. The agents' vision modules are always pretrained on a classification task, and different perceptual biases can be achieved via the different pretraining conditions explained above. Categories do not have to originate from language. Categories can also be formed through interactions with the world, and nonhuman animals as well as preverbal

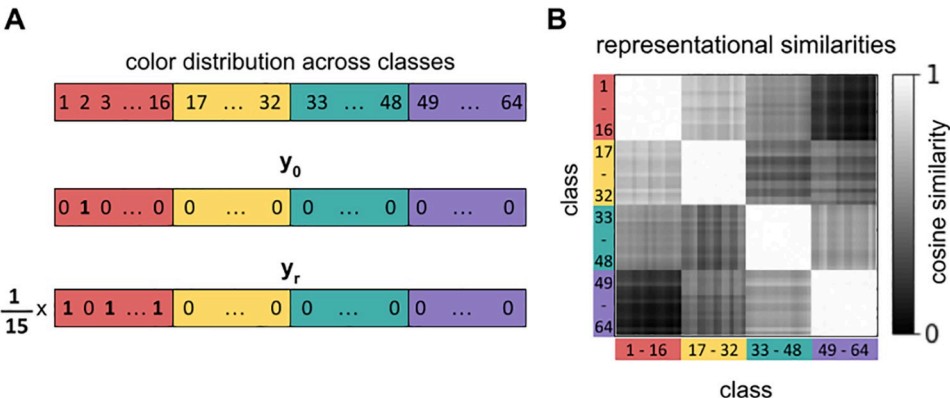

**Fig 2. Creating perceptual bias with relational label smoothing.** (A) Example of how the training targets (labels) are adapted to induce a COLOR bias. To generate a CNN with a COLOR bias, some of the target weight is spread across all other classes that *have the same color as the target object*. In our data set, there are 64 different object classes. The first sixteen classes comprise red objects (classes 1–16), followed by yellow objects (classes 17–32), turquoise objects (class 33–48), and purple objects (classes 49–64). For example, if the input image belongs to class 2 ("tiny red cylinder"), the usual target label, $\mathbf{y}_0$, is a one-hot vector where the entire weight lies on the true class index. The relational component, $\mathbf{y}_r$, spreads the target weight onto all other red objects. The target vector used for training is a weighted average of the original target and the relational component. Analogously, to introduce a scale/shape bias, some of the target weight is spread onto all other objects of the same scale/shape as the input object. (B) Representational similarity matrix for the COLOR CNN after training ($\sigma = 0.6$). Entries at position $(i,j)$ correspond to the average cosine similarity between the CNN activations for images of class $i$ and the CNN activations for images of class $j$ (based on the penultimate fully-connected layer). The white $16 \times 16$ blocks on the diagonal indicate that objects of the same color are perceived as very similar to each other.

human infants can learn categories [46]. Of course, these categories can still be lexicalized later on. The classification task is motivated by this ability to form categories through interactions with the world. While we do not explicitly model such interactions we assume they take place nonetheless. To study the influence of differences in perception on communication (Fig 3, top row), we train a sender and a receiver with fixed vision module weights on the communication game. The evolutionary analysis uses the same setup. Here, multiple games between sender-

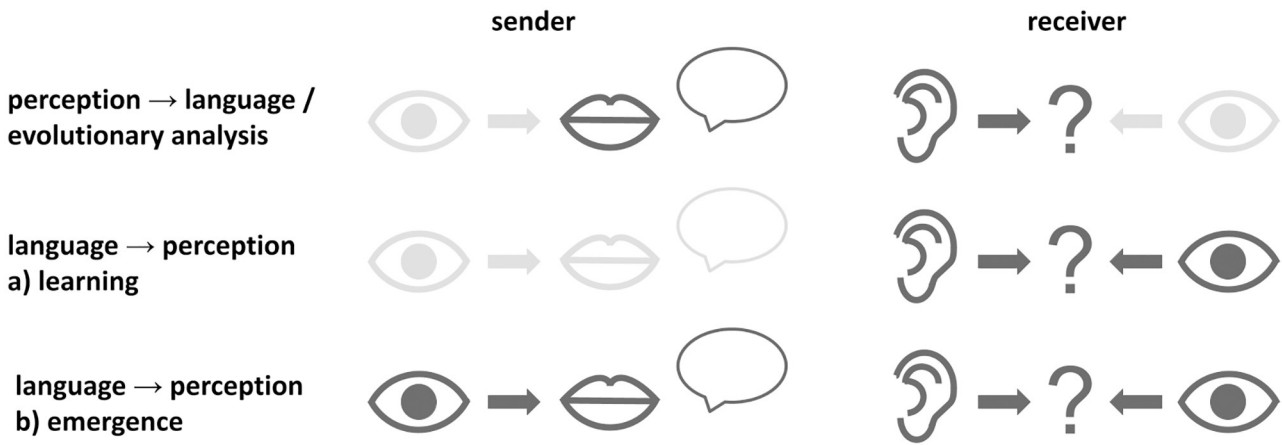

**Fig 3. Illustration of the training setups.** The vision module is represented by an eye, the language module by a mouth (sender) or an ear (receiver). The speech bubble represents the message, and the question mark the receiver's selection. Modules that are not trained, i.e. have fixed weights, are light gray. Modules that are trained are dark gray. Note that the vision modules in the two language emergence scenarios (center and bottom row) are trained on the communication game and simultaneously also on the original object classification task.

receiver pairs are used to approximate the communicative success of agent populations with different perceptual dispositions. To study the influence of language on perception, we consider language learning and language emergence (Fig 3, center and bottom row). In the language learning scenario, the language is fixed—using a trained sender—and only the receiver is trained, while in the language emergence scenario, both agents are trained. Importantly, in both scenarios, not only the language module but also the vision module is trained, such that changes in perception can occur. When learning to communicate, visual representations may adapt but they are still constrained by the functions of the visual system. In our case, this function is limited to object recognition (classification). To ensure that the agents' perceptual ability does not deteriorate to processing only aspects relevant to the communication game, training on the classification task used for pretraining continues. The loss function is generated by adding the classification loss and the communication game loss together.

**CNN pretraining.**    The CNN architecture consists of two convolutional layers with 32 channels, followed by two fully connected layers with 16 nodes, and a final softmax layer. The first convolutional layer is followed by a $2 \times 2$ max-pooling layer. For pretraining, we use stochastic gradient descent (SGD) with learning rate 0.001 and batch size 128, and train for 200 epochs. We set smoothing factors as high as possible while keeping the classification accuracy close to maximal. For the COLOR, SCALE, and SHAPE conditions, we use a smoothing factor of $\sigma = 0.6$. For ALL, the weight is distributed across more classes, which allows for a higher smoothing factor of 0.8. All networks achieve test accuracies >97%.

**Communication game.**    For most simulations, we use vocabulary size $|V| = 4$, message length $L = 3$, and $k = 2$ distractors. In principle, this allows agents to use a distinct symbol for each object and thereby to achieve maximal reward. As there are only a few distractors, agents may achieve relatively high rewards with suboptimal strategies. It is in the variation of such local solutions that we hope to identify linguistic differences that reflect perceptual biases and vice versa. We also run control experiments with a larger vocabulary size and more distractors, as well as control experiments changing the task-relevance of individual attributes. The agents minimize the negative expected reward, $-\mathbb{E}[r]$, and their trainable weights are updated using REINFORCE [47], which is a basic policy gradient algorithm. We train all agents using Adam with learning rate 0.0005 and batch size 128. Embedding and GRU layer each have a dimensionality of 128. We add an entropy regularization term [48] of 0.02 to sender and receiver loss to encourage exploration. The vision modules are initialized with the weights of the pretrained CNNs. When both agents are trained, training proceeds for 150 epochs, if only the receiver is trained (language learning) for 25 epochs.

## Evaluation

We are interested in the mutual influence between perception and language. Accordingly, we devise metrics to quantify perceptual biases as well as linguistic biases.

**Perception.**    Let $A$ = {*color*, *scale*, *shape*} be the set of object attributes, and $V_a$ all values that attribute $a \in A$ can take on, e.g., $V_{scale}$ = {*tiny*, *small*, *big*, *huge*}.

Given a set of inputs, *representational similarity analysis* (RSA) [49] measures the similarity between two representational spaces, by calculating the pairwise distances (in our case similarities) of input representations in either space and then correlating the two distance matrices. We use the analysis in two different ways. In the first case, RSA quantifies how well an agent's visual representations capture conceptually relevant attributes. Here, the two spaces under comparison are the space of the agent's visual representations generated by $v(\cdot)$, and a symbolic space of $k$-hot encoded attribute vectors ($k = |A| = 3$). In the second case, RSA quantifies the degree of perceptual alignment between an agent and its communication partner, and the two

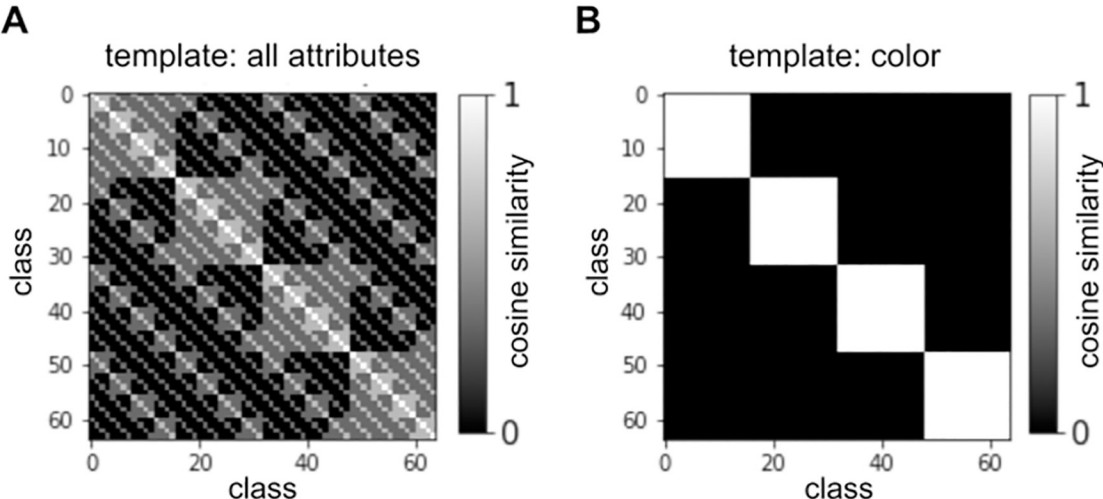

**Fig 4. Quantifying perceptual bias.** (A) Object similarities calculated from 3-hot encodings based on all three attributes. This template is used in the RSA calculation to measure how well conceptually relevant attributes are encoded. (B) Object similarities calculated from 1-hot encodings based on color value. This template is used to calculate $RSA_{color}$.

spaces under comparison are the two different visual representation spaces. In the first step, we extract $N = 50$ random example images for each object (class) and generate a representational similarity matrix (RSM) for each space under comparison, by calculating the pairwise cosine similarities between the corresponding representations, $sim_{cos}(r_i, r_j) = \frac{r_i^T r_j}{\|r_i\|\|r_j\|}$. Fig 2B shows an example of an RSM for a COLOR agent. In the second step, the actual RSA score is calculated as the Spearman correlation between the RSMs of the two spaces under comparison.

The RSA score with respect to the attribute template tells us how well differences in the underlying compositional object structure correlate with differences in the agent's visual representations. Fig 4A shows the RSM calculated from $k$-hot encoded attribute vectors, which serves as a ground-truth template. We can also use RSA to quantify whether agents can represent similarity relationships for some attributes better than for others. In order to do so, we replace the $k$-hot attribute vectors above by one-hot vectors encoding the values $V_a$ of a specific attribute $a$, and repeat the procedure for each attribute $a \in A$, resulting in separate RSA scores for color, scale, and shape. Fig 4B shows the color RSM template. Notice, that the RSA scores for individual attributes attenuate each other, as the agent's representations cannot simultaneously match all three templates. If one score is higher than the others, the agent represents one attribute at the cost of the others and is said to have a perceptual bias for that attribute. We denote the general RSA score (including all attribute values) by *RSA*, and the scores for a specific attribute by $RSA_a$.

**Language.** We use an information-theoretic evaluation to quantify the linguistic bias. Communicative success is based on what information about the target objects, *O*, the sender encodes in the messages, *M*, but also what information the receiver decodes from the messages to determine its object selections, *S*. Communicative success depends on both these factors, suggesting a three-way analysis, see Fig 5 (left), which would allow us to quantify the shared and distinct information between all combinations of objects, messages, and selections. However, in our experiments, the shared information between objects and selections is entirely predicted by the messages, since the receiver can only make selections based on message content (for details see S1 Appendix). Therefore, we can skip the object-selection interface, leading to

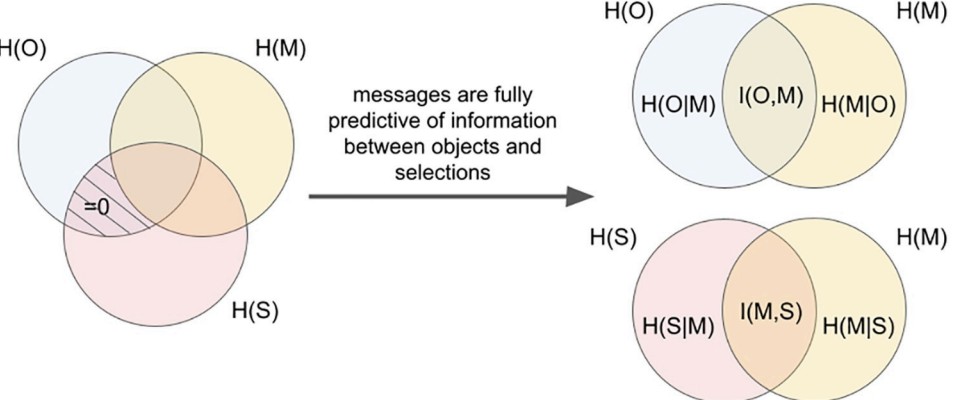

**Fig 5. Schema of the information in the target objects, *O*, the corresponding messages, *M*, and objects selected by the receiver, *S*.** *H* denotes entropy and *I* mutual information. The object-selection interface is entirely predicted by the messages as the mutual information between objects and selections given messages (shaded region on the left side) is zero. Therefore we can separate the analysis of sender (objects-messages) and receiver (messages-selections) as shown on the right. Note, the schema is not an actual set-theoretic representation and serves illustrative purposes only.

separate analyses of the relation between objects and messages, and messages and selections Fig 5 (right).

The mutual information between two random variables, $I(X, Y)$, measures how predictive these variables are of each other

$$I(X, Y) = H(Y) - H(Y \mid X) = H(X) - H(X \mid Y),$$

where $H(X)$ is the marginal entropy and $H(X|Y)$ the conditional entropy defined as

$$H(X \mid Y) = - \sum_{y \in Y, \ x \in X} p(y, x) \log \frac{p(y, x)}{p(y)} .$$

The conditional entropy indicates how much uncertainty about $X$ remains (on average) after learning $Y$. It turns out that, in all our experiments, the analysis of sender and receiver are symmetric in that $H(O \mid M) \approx H(S \mid M)$, $H(M \mid O) \approx H(M \mid S)$, and accordingly also $I(O, M) \approx I(M, S)$. Therefore we limit our analysis to the sender.

The conditional entropy, $H(O \mid M)$, quantifies the degree of uncertainty about the objects when knowing the messages that were sent. In reverse, to measure how much information about the objects is encoded in the messages, we can define an effectiveness score by

$$E(O, M) = 1 - \frac{H(O \mid M)}{H(O)},$$

with $E(O, M) \in [0, 1]$. To measure linguistic bias, we can define an effectiveness score for individual attributes. Let $O_a$ be the values of attribute $a$ for all objects, and $M$ the generated messages as above, then we can measure how much information about $a$ is encoded in the messages as $E(O_a, M)$. It follows, that

$$\overline{E(O_a, M)} = \frac{1}{|A|} \sum_{a \in A} E(O_a, M)$$

measures how well all conceptually relevant attributes are communicated. Unlike the RSA scores for individual attributes, $E(O_a, M)$, can be maximal for all attributes at the same time.

**Table 1. RSA between visual object representations and object attributes for each pretraining condition.**

|  | default | color | scale | shape | all |
|---|---|---|---|---|---|
| $RSA_{color}$ | 0.633 | 0.750 | 0.019 | 0.021 | 0.440 |
| $RSA_{scale}$ | 0.101 | 0.019 | 0.750 | 0.025 | 0.319 |
| $RSA_{shape}$ | 0.056 | 0.017 | 0.015 | 0.748 | 0.424 |
| $RSA$ | 0.439 | 0.437 | 0.437 | 0.442 | 0.675 |

Scores are calculated between object representations and $k$-hot attribute encodings, $RSA$ (bottom row), as well as for each individual attribute $a$, $RSA_a$.

## Results

This section presents analyses and results. At first, a validity check of label smoothing as a method to induce selective visual biases is performed. Then, each of the three questions under investigation is treated separately.

### Perceptual biases generated via label smoothing

**Relational label smoothing can systematically manipulate perception.** In order to test the validity of our manipulations, we check whether relational label smoothing induces the intended biases. As the agents' vision modules use object representations from the penultimate CNN layer, we quantify the biases for that layer using RSA. t-SNE plots [50] and pairwise class similarities of object representations can be found in S1 and S2 Figs. Table 1 shows the RSA scores for each of the five pretraining conditions. Surprisingly, the DEFAULT CNN represents differences in color values much more accurately than differences in other attributes. This inherent color bias may be due to the networks' direct access to color information via the RGB channel input [51]. COLOR, SCALE, and SHAPE networks mostly capture differences in the respective attribute. The ALL network represents differences in all three attributes, which can be seen from relatively high RSA scores per attribute, as well as a higher overall RSA score. Note, maximum values per attribute are smaller than in the other conditions due to mutual attenuation. In conclusion, by default, object representations extracted from CNNs are biased towards representing color information but relational label smoothing can shift this bias to other attributes as well as improve coverage of the entire input topology.

### Influence of perception on language

To quantify the influence of different visual biases on emergent communication, we trained agents with different visual biases (and fixed vision module weights) on the communication game. For all CNNs (DEFAULT, COLOR, SCALE, SHAPE, ALL) we trained a sender-receiver pair where both agents used the same vision module and thus had the same bias. In addition, to evaluate the impact of sender versus receiver bias we ran experiments combining a DEFAULT receiver with each type of sender, and combining a DEFAULT sender with each type of receiver. We conducted twenty runs per agent combination. All agents learned to play the game, with mean test rewards ranging between 0.914–0.968 (details about the agents' performance follow later in this section).

**Perceptual biases systematically shape emergent language.** We begin by analyzing the effect of perceptual biases on emergent language when both agents have the same bias. We use the effectiveness score to measure how much information about specific attributes is contained in the messages. The results for each type of bias and each attribute are shown in Fig 6A. The five blocks on the $x$-axis show the perceptual bias conditions, with each bar representing one

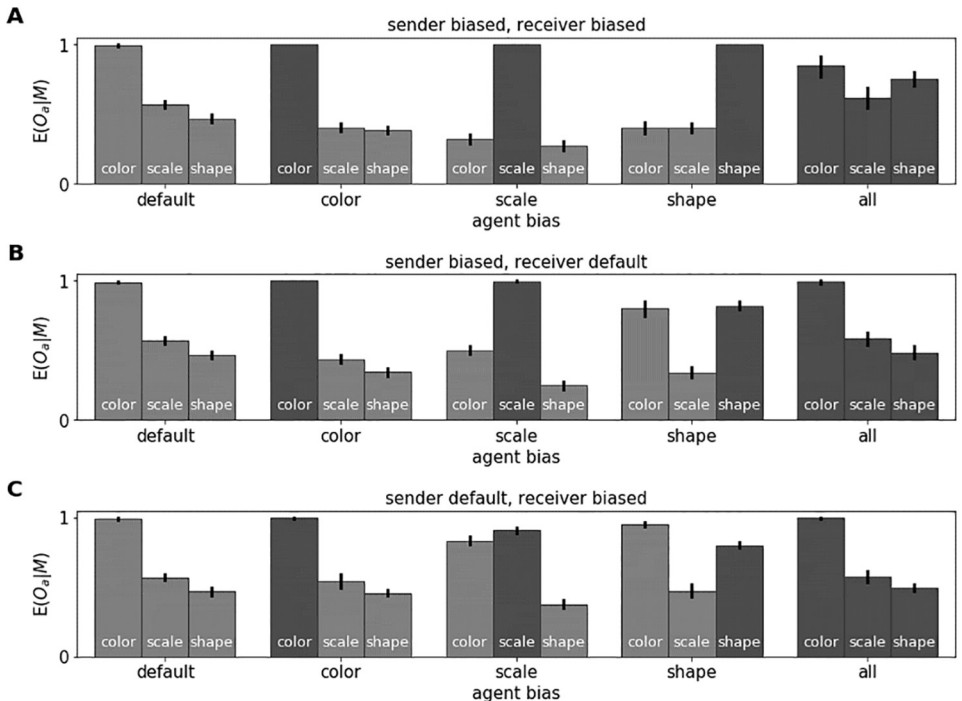

**Fig 6. Effectiveness per attribute for different pairings of senders and receivers.** Pairings are (A) biased sender and biased receiver, (B) biased sender and DEFAULT receiver, and (C) DEFAULT sender and biased receiver. The *x*-axis shows the agents' perceptual biases. The bars are labeled with the attribute *a* used for calculating $E(O_a|M)$, with attributes enforced via label smoothing in dark gray. We report means and bootstrapped 95% CIs of twenty runs each.

of the three attributes. In the DEFAULT condition (left) the messages are strongly grounded in object color, which can be attributed to the inherent color bias of the DEFAULT CNN. Agents with a color, scale, or shape bias (central three blocks), ground their messages to a large extent in the attributes they have a perceptual bias for. Overall, the effectiveness across conditions is significantly higher for biased attributes ($M = 0.868$) than unbiased attributes ($M = 0.468$), as indicated by a bootstrapped 95% confidence interval (CI) for the difference in means of [0.355, 0.444]. Qualitatively, the observed patterns prevail also if the vocabulary size and the number of distractors are increased, both of which encourage the agents to communicate more information about each attribute (see S2 Appendix). It seems that if agents are good at perceiving object similarities along specific dimensions, they prefer to communicate these dimensions over others.

**Sender bias is more influential than receiver bias.** Effectiveness scores for varying the sender bias in combination with a DEFAULT receiver are shown in Fig 6B, and for varying the receiver bias in combination with a DEFAULT sender in Fig 6C. The results for DEFAULT from part (A) are repeated as a reference. Comparing part (B) to part (A) of the figure, and singling out the effects of color, scale, and shape biases, biasing only the sender has similar effects as biasing both agents. For each of these biases, the language is grounded largely in the corresponding attribute. Still, the color bias of the DEFAULT receiver leads to an increase in color effectiveness when the sender itself does not have a color bias. Comparing (C) to (B), also a receiver bias is carried over into the emergent language, even though its influence is weaker and the color bias of the DEFAULT sender dominates. We calculate the mean absolute difference (MAD) between the average effectiveness scores in (B) and (A), as well as (C) and (A), for COLOR, SCALE, and

**Table 2. Training rewards, test rewards, and average effectiveness across attributes for sender-receiver pairs with the same bias.**

|  | default | color | scale | shape | all |
|---|---|---|---|---|---|
| train reward | 0.956 ± 0.003 | 0.928 ± 0.008 | 0.910 ± 0.006 | 0.937 ± 0.008 | **0.968** ± 0.004 |
| test reward | 0.959 ± 0.003 | 0.929 ± 0.009 | 0.914 ± 0.007 | 0.939 ± 0.008 | **0.968** ± 0.004 |
| $\overline{E(O_a, M)}$ | 0.676 ± 0.013 | 0.596 ± 0.015 | 0.532 ± 0.016 | 0.600 ± 0.020 | **0.738** ± 0.017 |

Reported are means and bootstrapped 95% CIs calculated from twenty runs per condition. The best values across conditions are highlighted.

SHAPE condition, to quantify the relative influence of biasing one versus both agents. The imbalance between sender and receiver bias is reflected in a higher MAD for biased receivers (0.194) than biased senders (0.103). Looking at the ALL condition, an interesting pattern emerges. If both agents have an ALL CNN as in (A), the message information is more evenly distributed across all attributes than in the DEFAULT condition. However, if either of the agents uses a DEFAULT CNN, as in (B) or (C), this effect is reversed and the messages are mostly grounded in color, which is likely because the "flexible" ALL agent adapts to the inherent color bias of the DEFAULT agent. In line with this interpretation, the MAD between average effectiveness scores in ALL condition and DEFAULT condition is very small, both when the sender is biased (0.012) and when the receiver is biased (0.013). In sum, perceptual biases of both sender and receiver are reflected in the emergent language, but due to the asymmetry of communication, the sender bias is more influential. Further, agents that rely strongly on all conceptually relevant object dimensions for perceptual categorization can flexibly adapt their language to suit communication partners with more narrow perceptual discrimination abilities.

**Perception of relevant similarity relationships improves communication.** Table 2 displays the training rewards, test rewards, and average effectiveness across attributes for all five conditions (sender and receiver biased). Results for pairing biased with DEFAULT agents can be found in S1 Table. The mean test rewards range between 0.914–0.968 across all conditions, at a chance level of 0.33. We are particularly interested in the ALL versus DEFAULT comparison, so whether sharpening the agents' perception with respect to conceptually relevant dimensions improves emergent communication in comparison to default processing. According to all three metrics, ALL agents achieve the best values, and DEFAULT agents the second-best values. The strong perceptual bias for individual attributes seems to bias the communication to a degree that is harmful to performance. Still, the differences between ALL and DEFAULT are significant based on the bootstrapped 95% CIs for the difference in means with respect to training rewards ([0.007, 0.017]), test rewards ([0.005, 0.014]), and average effectiveness ([0.040, 0.083]). The higher average effectiveness in the ALL condition suggests that enforcing conceptually relevant similarities helps the agents to overcome categorization biases, such that they can better communicate all relevant attributes—instead of forming semantic categories based on individual attributes—and as a consequence achieve higher performance.

## Influence of language on perception

To study the influence of different linguistic biases on visual perception, we considered a language learning and a language emergence scenario. For the language learning scenario, we used the trained senders from the agent pairs above (where both agents have the same bias) and trained DEFAULT receivers to learn their language. For the language emergence scenario, we ran experiments combining a DEFAULT receiver with each type of sender, and combining a DEFAULT sender with each type of receiver. We conducted ten runs per scenario and agent

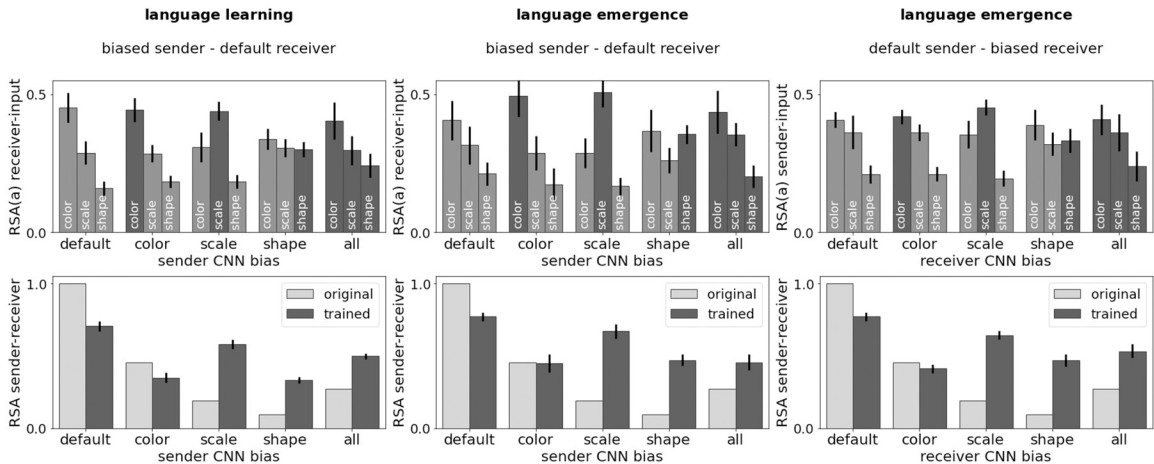

**Fig 7. Influence of linguistic biases on perception.** Shown are the effects of language learning and language emergence on a DEFAULT agent, when paired with agents of different visual bias conditions. The left column covers the language learning scenario with a DEFAULT receiver, the central column the language emergence scenario with a DEFAULT receiver, and the right column the language emergence scenario with a DEFAULT sender. In the language learning scenario, the sender's weights (and therefore also the language) are entirely fixed. In the language emergence scenario, both agents are trained and the language emerges. The visual bias of the communication partner is shown on the *x*-axis. The top row shows the RSA scores between the DEFAULT agent's visual representations and each object attribute—indicated by the bar label—after training. Attributes that were enforced to create the visual bias of the communication partner are dark gray. The bottom row shows the RSA scores between the visual representations of the DEFAULT agent and those of its communication partner before (light gray) and after (dark gray) training. Reported are means and bootstrapped 95% CIs of ten runs each.

combination, with mean test rewards ranging between 0.919–0.973 (for details about training and test rewards see S3 Fig).

**Linguistic biases influence perception.**   In the language learning scenario, the language was fixed and learned by the receiver. Fig 7, top left, shows that the linguistic biases clearly influence the agent's perception: if message content is biased towards a specific attribute—as in the DEFAULT (color attribute), COLOR, SCALE, and SHAPE condition—the agent learns to better represent visual differences for this attribute. As the DEFAULT receiver starts out with a perceptual color bias (see Table 1), changes in visual perception are most clearly visible in the SCALE and SHAPE conditions, where the color bias is reduced, and scale or shape bias increases. Looking at the RSA scores between the sender's and the receiver's visual object representations (Fig 7, bottom left) we find that unless both agents start out with a color bias (DEFAULT and COLOR condition) the scores increase, so the receiver's representations adapt to those of the sender. The center and right columns of Fig 7 visualize the same analysis results for the language emergence scenario, once for a DEFAULT receiver paired with senders from different conditions (center), as well as for a DEFAULT sender paired with receivers from different conditions (right). The exact same qualitative patterns as in the language learning scenario emerge, with differences in amplitude suggesting that the receiver is more affected by the sender's bias than vice versa. The agents' biases are passed on through language, even if there is no fixed linguistic protocol to begin with.

**Communication can improve perception of relevant similarity relationships.**   Color, scale, and shape information is relevant for the communication game. Therefore, it seems plausible that playing the game could improve visual object representations with respect to these attributes. Fig 8 shows the RSA scores of a DEFAULT agent after training in the language learning scenario (left), and the language emergence scenario as receiver (center) or sender (right). The CNN type of the communication partner is color-coded. Indeed, compared to the

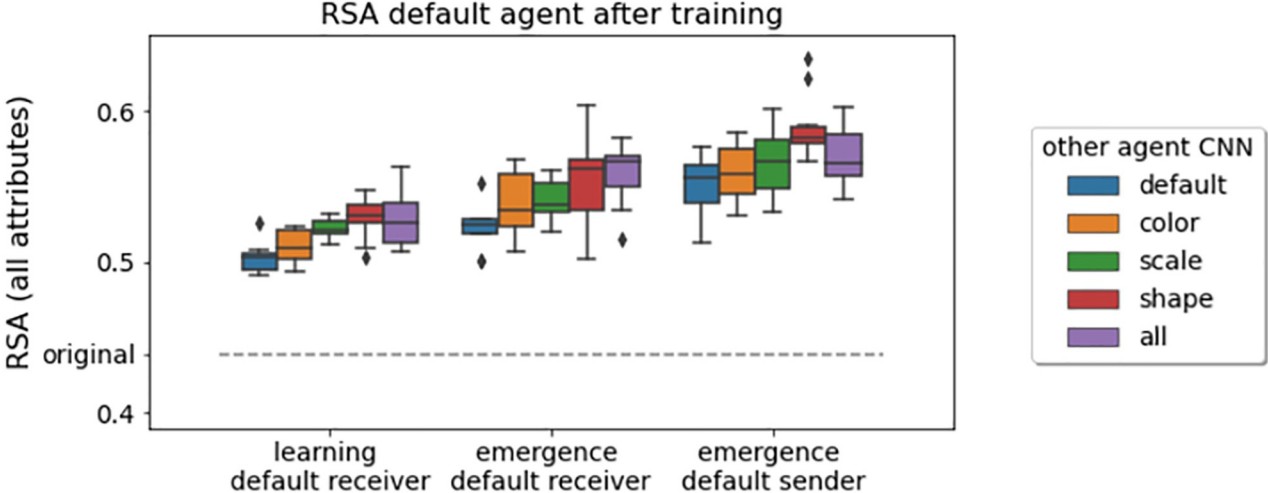

**Fig 8. RSA scores between symbolic object representations (*k*-hot attribute vectors) and neural object representations in the agent's vision module.** Shown are the scores for the DEFAULT agent after training, for different communication partners, across ten runs each. For the language learning scenario, the DEFAULT receiver is shown (left). For the language emergence scenario, the DEFAULT receiver (left) and the DEFAULT sender (right) are shown. The dashed line indicates the RSA score of the DEFAULT CNN—so the agent's vision module—before training.

original RSA score, regardless of the scenario and the bias of the communication partner, the CNN of the DEFAULT agent better accounts for differences in the conceptually relevant attributes. The representational grouping of objects based on the inherent CNN color bias is reduced by playing the communication game.

We further analyzed the influence of scenario (learning, emergence—DEFAULT receiver, emergence—DEFAULT sender) and communication partner bias (DEFAULT, COLOR, SCALE, SHAPE, ALL) by looking at the bootstrapped 95% CIs for the differences in means. Mean RSA scores are lowest in the learning scenario ($M = 0.518$). They are higher in the emergence scenario with a DEFAULT receiver ($M = 0.543$), with a CI of [0.017, 0.033], and even higher for the emergence scenario with a DEFAULT sender ($M = 0.567$), with a CI for the two emergence scenarios of [0.014, 0.033]. Agents in the language emergence scenarios learn object representations that better reflect the underlying object structure compared to agents in the language learning scenario, with a stronger effect for the sender than the receiver. Thus, it is beneficial, if both agents can adapt their perceptual processes to the game. As the sender dominates the emerging protocol (see above), its visual representations might adapt more strongly to the task. With respect to differences in communication partner bias, we were particularly interested in which communication partners can increase the RSA score compared to a DEFAULT partner ($M = 0.525$ across scenarios). In pairwise comparisons with the DEFAULT partner, a partner with a SHAPE bias leads to the strongest improvement ($M = 0.558$, $CI = [0.017, 0.047]$), followed by ALL ($M = 0.552$, $CI = [0.014, 0.040]$), then SCALE ($M = 0.543$, $CI = [0.005, 0.030]$), and finally COLOR does not seem to yield a significant improvement ($M = 0.535$, $CI = [−0.003, 0.022]$). The DEFAULT agent is good at representing differences in object colors, and bad at representing differences in both scale and shape information, with the largest deficit for shape (see Table 1). It seems that talking to SHAPE or ALL agents, which are good at representing shape information, can help overcome the shape deficit, therefore leading to the strongest improvements. Similarly, communication with a COLOR agent does not stimulate the agent to adapt its representations, as the preferred structure based on color values is mutual.

Overall, adapting visual perception for a downstream communication task (while staying true to the original classification objective) improves the visual representation of task-relevant aspects of the environment—in our case the three object-defining attributes. The improvement is stronger if the communication partner is good at representing aspects for which the agent has a deficit.

**The role of classification.** The agents' vision modules are trained for classification and communication at the same time. The classification task is used to simulate that the visual representations have other purposes apart from informing communication. We ran additional control simulations without the classification task, to understand its influence on the results above. A detailed description of methods and results can be found in S3 Appendix. The main finding can be confirmed also without classification: If message content is biased towards a specific attribute—because it is predetermined (language learning) or arises through a visual bias of the communication partner (language emergence)—the DEFAULT agent learns to better represent visual differences for this attribute. Still, the classification loss has a moderating effect on the RSA scores as it constrains the visual representations to capture differences between the values of all attributes regardless of linguistic bias. In other words, it keeps the vision module from only representing information that is relevant to the communication game. As the agents discriminate between fewer objects in communication than in classification (communication is less optimal than classification), playing the reference game does not improve the visual representations, i.e. the general RSA score, without the classification loss.

## Evolutionary analysis

In the preceding analyses we studied how perceptual representations are affected by language use. Here, we take this idea to an extreme by analyzing whether specific perceptual representations (biases) are more likely to result from within- or cross-generational adaptation processes based on their aptitude for communication. For this purpose, we use the static solution concept of evolutionary stability from evolutionary game theory [28]. This solution concept assumes a large, homogeneous population where agents are randomly paired to play a game of interest. Based on the reward (or payoff) structure between different types of agents, it can be decided whether a population of a certain type can be invaded by an alternative type. In a two-player symmetric game, type $t$ is evolutionary stable, if agents of any mutant type $t'$ achieve less reward playing with an agent of type $t$ than two agents of type $t$ playing with each other, $r(t, t) > r(t', t)$. If there is a competing type $t'$, such that $r(t', t) = r(t, t)$, $t$ is still evolutionary stable if $r(t, t') > r(t', t')$.

While the concept of an ESS has first been introduced in the context of biological evolution, it is useful also for analyzing the stable rest points of non-biological evolutionary optimization processes The latter is made possible by the fact that ESSs are the (locally) asymptotically stable rest points of the replicator dynamic [52, 53]. The replicator dynamic, in turn, is a rather encompassing high-level formalization of a wide variety of agent-internal optimization processes, be they cross-generational as in cultural evolution or (asexual) reproduction [54], or within-generational as in imitation-based dynamics [54, 55] or simple forms of reinforcement learning [56].

**Enhanced perception of relevant features is evolutionary stable.** In our case, the game of interest is the reference game, and the different types are given by different perceptual biases. We assume that agents in the population can act as both sender and receiver. Accordingly, the rewards for two communicating agents with biases $t$ and $t'$ are calculated by averaging the rewards of a $t$-sender paired with a $t'$-receiver and a $t'$-sender paired with a $t$-receiver. This is also known as *symmetrizing* the game [57, Section 3.4]. Because the training process

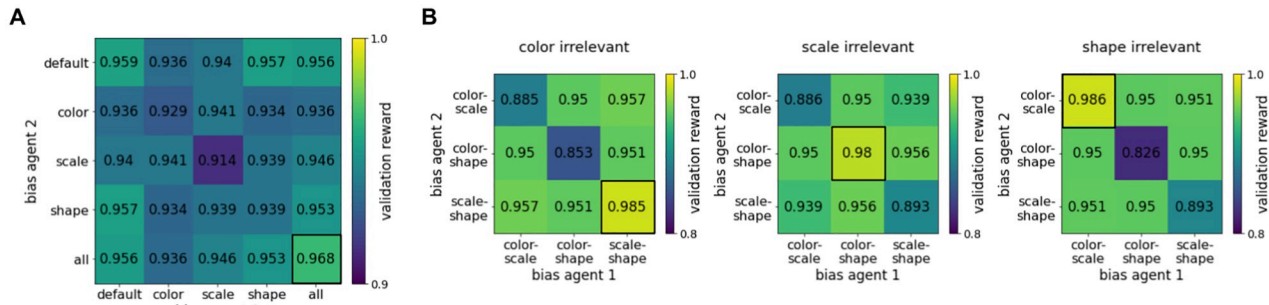

**Fig 9. Mean reward on the test set for two agents of different bias types communicating with each other.** For each sender-receiver combination, we ran twenty simulations. To obtain the average reward for an agent of bias type $t'$ communicating with an agent of bias type $t$, we average the rewards of the combinations $t'$-sender/$t$-receiver and $t$-sender/$t'$-receiver, hence the matrices are symmetric. We highlight the results for the combinations where both agents are biased towards all relevant attributes. (A) shows the mean test rewards for agents with $t'$, $t \in$ {DEFAULT, COLOR, SCALE, SHAPE, ALL} in the basic reference game where all attributes (color, scale, shape) are relevant. (B) shows the mean test rewards for agents with mixed biases $t'$, $t \in$ {COLOR-SCALE, COLOR-SHAPE, SCALE-SHAPE} for reference games where out of the three attributes either color (left), scale (center), or shape (right) is not relevant.

and the agents' policies are stochastic, the reward for an interaction between two bias types is approximated by averaging across multiple runs. Fig 9A shows the reward matrix for all bias combinations averaged across twenty simulations for each sender-receiver pair. Judging from the average rewards, the DEFAULT and ALL conditions form the only evolutionary stable biases. Pairwise comparisons between the CIs in each matrix column reveal that only the evolutionary stability of the ALL bias is significant. Thus, the ALL bias prevails in an optimization process for communicative success.

**Eliminating potential confounds of task-relevance as evolutionary drive.** ALL agents achieve higher rewards than other agents. Intuitively, this is the case because the ALL condition enforces task-relevant attributes. If object color was not relevant to the game, enforcing color similarities should not increase performance, and a color bias should not evolve. However, the advantage of ALL agents could be due to other factors. We noted above that, based on the nature of the reference game, the conceptually relevant (i.e. class-defining) attributes correspond to the attributes that are relevant for successful communication. To achieve perfect performance, all conceptually relevant attributes must be communicated, such that the receiver can identify the target unambiguously against different distractors. ALL agents could therefore achieve higher performance because they are biased towards class-defining attributes rather than task-relevant attributes; or, simply because more attributes are enforced than in the other conditions, which might improve representational structure.

To exclude these alternative explanations, we ran a set of control simulations. We created different mixed-bias conditions, where similarities for two out of three attributes were enforced during perception-pretraining (COLOR-SCALE, COLOR-SHAPE, SCALE-SHAPE). To ensure that the bias strength for enforced attributes is high and approximately equal within and across types, as well as that the bias strength for unenforced attributes is approximately zero, we conducted a grid search across different smoothing factors and weightings between the two enforced biases (for details see S4 Appendix). In addition, we designed reference game variants, where always one of the three object attributes is not relevant (color irrelevant, scale irrelevant, shape irrelevant). E.g., if object color is irrelevant, sender and receiver target may have different colors and still yield maximal reward, while scale and shape must be the same, see Fig 10. By training combinations of mixed-bias agents on these games, the set of attributes relevant

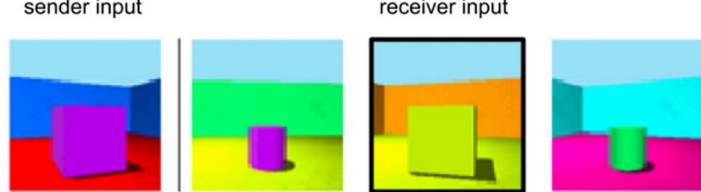

**Fig 10. Example inputs if object color is irrelevant in the communication game.** The receiver target is marked by a black box. S4 Fig shows examples of sender and receiver inputs for each game variant (color irrelevant, scale irrelevant, shape irrelevant).

to pretraining is disentangled from the set of attributes relevant to communication, while the number of enforced biases is constant across agent types.

Fig 9B shows the resulting reward matrices (for an analysis of the linguistic biases see S5 Fig). In each game variant, agent types with a bias for task-relevant attributes form the only evolutionary stable population. Particularly low performances arise when both agents have the same mismatching bias (low values on the diagonal) because, in that case, the agents' bias does not encourage communication about the respective "missing" attribute. E.g., if both agents have a COLOR-SCALE bias, introducing shape information into the conversation is more difficult than if one agent has a COLOR-SHAPE bias. The matrices further show that representations which are biased towards task-relevant attributes will win against any alternative homogeneous bias. In conclusion, there might be optimization pressure towards representations that accurately capture the relationships between objects, in terms of features that are environmentally relevant.

## Discussion

We proposed that communication games with deep neural network agents can be used to study interactions between perception and emergent communication. Based on systematic manipulations of visual representations and communication protocols, we made the following main observations: 1) biases in either modality are reflected in the other, 2) communication improves the perception of task-relevant attributes, and 3) enforcing accurate representation of task-relevant attributes improves communication—to a degree that specialization of the perceptual system to the linguistic environment could accrue.

Multi-agent communication games account for the interactive and grounded nature of communication. Reinforcement learning (RL) presents a natural framework for modeling learning in these games. Utterances are treated like actions: they are grounded in the environment and driven by objectives. Machine learning models trained on language in isolation—typically under (self-)supervision—have achieved impressive results on various natural language processing tasks by capturing statistical patterns from large corpora [58–60]. However, lacking a grounded shared experience, these models cannot address deeper questions about communication and meaning [3].

### Influence of perception on language

The first set of analyses investigated the influence of visual perception on emergent communication. We found that semantic category formation was largely shaped by perceptual similarity relationships. In human cognition, the idea that many concepts are characterized by perceptual properties is uncontroversial. For example, objects that are grouped under the same psychological concept often have similar shapes [61]. The conceptual structure of

the world in our reference game is predetermined: objects are defined by color, scale, and shape, each being equally important. Still, the agents group together several concepts under a single label based on perceptual similarity, which means the emerging protocol is suboptimal. They even do so when the message space and the number of distractors are increased (see S2 Appendix). Recently, it was shown that neural network agents playing a color discrimination game develop *efficient* communication, in the sense that they reach maximum accuracy for a given language complexity, and that—as in human color-naming systems—low complexity is preferred [4]. We assume a similar effect in our simulations. The agents develop accurate but simple protocols, and reductions in complexity are achieved by grouping different objects under the same label based on perceptual similarity. We further showed that increasing the perceptual sensitivity for features that are relevant to the communication game debiases communication and improves performance. In line with the above interpretation, it could be that agents with better adapted representational spaces find solutions with higher complexity and accuracy, while still optimizing the trade-off between the two.

These results are also relevant from an engineering perspective. A lot of the existing research in language emergence is focused on developing setups that foster the emergence of communication protocols sharing desirable properties with natural language. The role of how agents perceive and represent the world is mostly ignored [43]. However, we not only show that perceptual biases directly influence the emerging protocol but also that they are present in default setups. We find that the organization of pixel inputs into dedicated color channels makes color information more easily accessible than other object information, which leads to a color bias in communication. Neural networks process visual information differently from humans in many ways. For example, they are susceptible to adversarial attacks [62] and lack useful learning mechanisms observed in children [63]. We think that language emergence research can profit from taking into account the effects of differences between human and machine perception. Moreover, we show that agents' performance can be improved by developing representational similarity relationships that are based on task-relevant dimensions, rather than using out-of-the-box pretrained networks.

## Influence of language on perception

The second set of analyses studied the influence of (emergent) communication on visual perception. We found that categories established by the communication protocol modulate representational similarities to better reflect this categorical structure, by increasing the similarity between objects that are grouped together under the same expression. It has been shown that learning new color categories (through a perceptual task) induces categorical effects on color discrimination similar to those of natural color categories [64]. These results suggest that cross-language differences in perceptual representations may arise as a result of learning linguistic categories, as simulated in our experiments. Besides, we observed that perceptual sensitivity increases for features that are relevant in the communication game and therefore affect the agents' objective. The need to discriminate between features, for communication to be successful, can disentangle their visual representations. This increase in sensitivity occurs even though the exact same features are also relevant in the pretraining classification task. A related effect has been observed in a visual search task. Although there is a baseline effect of conceptual categories on visual processing, this effect increases if the target category is labeled [65].

Both these observations have been made in earlier simulations. Harnad, Hanson, and Lubin, showed that neural networks trained on a supervised classification task show effects of

categorical perception, in that a continuous input dimension is warped in the network representations to increase within-category similarity and decrease between-category similarity [66]. Later, Cangelosi and Harnad compared agents that learned categories from sensorimotor interaction with the world ("sensorimotor toil") to agents that could additionally learn from communication signals ("symbolic theft") [67]. Sensorimotor interaction, comparable to our pretraining classification task, warped the agents' representational similarity space but supervised learning of symbolic object descriptions warped these similarity spaces even further, leading to increasingly categorical perception. Our work extends these computational approaches. We model how a representation space can restructure itself to reflect a categorical partition of a comparatively complex input space, based on communicative interaction rather than supervised learning.

Modeling a communication scenario has the advantage that we can study interactions between communication partners who conceptualize the world differently. Because the emerging language is shaped by the perceptual biases of both agents, and in turn shapes their perceptual biases, the agents' representations become aligned through communication. Comparable effects have been found in empirical studies. Category structure aligns between people who play a reference game [68], and more generally between people who assign novel labels to stimuli with the goal to coordinate [69].

These analyses, too, have implications for engineering-driven research. Backpropagating the learning signal from the communication game through the vision module of the agents improves their ability to represent and discriminate between relevant features, which might be useful for downstream tasks other than communication. It also provides a way to align perceptual representations of different agents, which can be particularly useful if one agent can thereby correct specific perceptual deficits of the other agent.

## Evolutionary analysis

Finally, the evolutionary analysis showed that accurate perception of environmentally relevant aspects constitutes a functional advantage. Related results have been found in experiments with robots playing a color naming game [37]. Robots that could adapt their categories to the task performed better than robots starting out with the same, but fixed category structure. Most likely, representational structure in humans is optimized to accommodate environmentally relevant conceptualizations as well [70, 71]. In our simulations, communication was the only task performed by the agents. Representational structure in humans, however, is shaped by various environmental pressures. Our results do not indicate that perception only adapts to optimize communication, but rather that communication (as a means to exchange information about relevant aspects of the environment) may constitute one of these pressures.

Whether language could have influenced the brain, and therefore also visual perception, through biological evolution is highly debated. A major problem lies in the fact that it is difficult to estimate the relative change of perception during the evolution of language. The (macaque) monkey visual system is often and successfully taken as a model for the human visual system. A mainstream view is that the two visual systems share many characteristics but are not identical [72, 73]. Furthermore, it is uncertain when language emerged [74]. However, it has been argued that—evolutionarily young and variable—language is rather shaped by the—evolutionarily old and stable—brain than vice versa [75]. While we abstain from claims about the time scales of the analyzed optimization process, it seems more likely that language-guided adaptations of visual representations happen within the lifetime of an individual.

Stable state analysis is a static solution approach to evolutionary games. It can identify whether a given population will remain at a certain state but does not explain how a population

arrives at that state. The latter question can be answered by dynamic approaches, which apply an explicit model of the optimization process. A prominent example is the *replicator dynamic*, originally defined for a single species by Taylor and Jonker [52] and named by Schuster and Sigmund [76]. Thus, evaluating the probability that a randomly initialized population develops perceptual representations that match communicative needs would require the use of dynamic models.

## Flexible-role agents and populations

In the original Lewis game, there are two agents with fixed roles (sender and receiver), two world states, and two actions. In the theoretical analysis of signaling games, it has been of general interest how the agents' behavior changes under variations of this simple case [77]. Like the original Lewis game, our reference game involves two agents with fixed roles. To make sure that our results do not only pertain to this special case, we ran additional simulations with more agents and flexible-role agents. In particular, we separately tested an extension to flexible-role agents and an extension to a 4-agent game (two senders, two receivers). We repeated the analyses above for the DEFAULT, SCALE, and ALL conditions, as these conditions cover the main manipulations of enforcing no bias, a bias for a single attribute, or a bias for all attributes. Details about methods and results can be found in S5 Appendix (4 agents) and S6 Appendix (flexible-role agents). At least for these two extensions, we can establish the same main results as for the fixed-role, 2-agent game. While many more variations are conceivable, our findings seem to reflect general aspects of language-perception interactions in multi-agent communication.

## Limitations

Combining communication games with deep learning to study interactions between language and perception (and possibly other areas of cognition) is a novel approach. As a first implementation, the proposed setup tries to strike a balance between the flexibility of modern DNNs and experimental control. Our images and categories fall clearly short of the visual complexity of the world. However, using objects that are composed of a fixed set of attributes and attribute values has several advantages. We can introduce selective visual biases via relational label smoothing, and we can quantify and compare visual and linguistic biases with respect to these attributes.

Our model also greatly simplifies the functionality of visual perception. Our agents use their vision modules to generate representations that can be used for classification and communication. The visual brain, in contrast, performs a multitude of functions, each of which imposes organizational and representational constraints. In particular, visual perception requires an (implicit) understanding of sensorimotor contingencies as it informs and is informed by motor action [78]. Hence, unlike our model, the visual system represents information that is irrelevant to categorization or communication. As a consequence, our results likely overestimate the effects of language on perception. In addition, without a significant increase in architectural and functional complexity, an analysis of the penetration depth of language into visual representations (high-level attentional selection mechanisms vs. dynamic retuning of receptive fields of primary sensory neurons) does not warrant conclusions about the human visual system. Empirical studies show that the effects of language on vision are dynamic and task-dependent. For example, in color discrimination tasks, categorical effects are observed for naive but not trained observers [79], and sometimes only in the presence but not in the absence of verbal cues [21]. Future work could study these more nuanced effects by using more complex vision modules.

## Outlook

Our vision modules are CNNs trained on classification. Thereby, they rely on the same principles—albeit being much simpler—as state of the art models of vision [80, 81]. Still, there are many ideas on how correspondence between artificial and biological neural networks can be further improved by changing architectures, learning algorithms, input statistics, or training objectives [82, 83]. As a relatively minor change, training on superordinate or both superordinate and basic labels, rather than on subordinate labels as is typically the case, makes visual representations more robust and more human-like [84]. Note that information about taxonomic relationships can also be encoded in the training labels directly using the (hierarchical) relational label smoothing method presented here. An example of an architectural change are recurrent CNNs, which include not only bottom-up but also lateral and top-down connections. Including recurrence improves object recognition, especially under challenging conditions [85], and is required to model the representational dynamics of the visual system [86]. As an example of a change in objective, an *embodied* DNN agent has been shown to learn sparse and interpretable representations through interactions with its simulated environment [87]. In addition to scaling our experiments to more complex input data and deeper networks, future work could draw on these exciting developments to better capture the functional and architectural constraints on the visual system. The resulting models could be used to investigate how the effect of communication on perceptual representations changes under these additional constraints.

This paper set out to explore mutual influences between language and (visual) perception in multi-agent communication. But language interfaces with other areas of human cognition as well. The embedding of language in general cognition is evident in everyday language use. For instance, in understanding a written text, we are able to recruit from memory the right background assumptions to make the text coherent [88]. This can, among others, be observed in bridging inferences. Upon reading "They had a barbecue. The beer was warm.", we can conclude that the beer was part of the barbecue. Another salient example is attention. While we may share a basic attention mechanism for dealing with the non-linguistic world, having a language to "bridge minds" will likely lead to fine-tuning and, in fact, align our attentional mechanisms. Think about saying "Wow!" or adding "surprisingly". These so-called mirative markers convey surprise [89], thereby telling the audience what we expected, but also what we pay attention to. Essentially, every statement about the world conveys meta-information about what the speaker finds newsworthy in the first place. On a basic level, also the role of attention or memory could be studied with our setup, for example by using neural network agents with attention mechanisms [90] or external memory [91]. In general, due to the versatility of both deep learning architectures and communication games, their combination forms an excellent testbed for various language-related interface problems.

Our experiments go beyond analyzing effects *on* emergent communication. They also account for the reverse direction, i.e. how language shapes other domains. Such Whorfian effects are widespread; apart from visual perception they have, for example, been observed in motion, spatial relations, number, and false belief understanding [92]. In fact, it seems likely that all interfaces between cognition and language are mutually adapted towards optimal interaction in the environments we face [93], such that language can guide the acquisition of cognitive representations from experience, and in turn, can be used to structure and exchange these experiences [94]. In a neural network agent, linguistic feedback can be backpropagated into any module that may be considered adaptive to language use. As illustrated by our analyses, language emergence games can address adaptions within and across generations. Future

research could use the presented framework to improve our understanding of language in relation to general cognition, from its origins to its cultural and potentially genetic evolution.

## Supporting information

**S1 Fig. Two-dimensional t-SNE plots of the visual object representations in the penultimate CNN layer for each pretraining condition.** The four color and scale values are given by the four marker colors and marker sizes, while the following mapping from object shape to marker shape is used: (cube, sphere, cylinder, ellipsoid) → (square, circle, square cap (⊓), rhombus (◇)). t-SNE embeddings were calculated on a data subset of 100 random examples per class (6400 data points) using a perplexity of 100, and 2000 iterations. Plotted are the embeddings for 5 random examples per class. In the DEFAULT and COLOR conditions, clusters form around color values, in the SHAPE condition around shape values, and in the SCALE condition around scale values. The complex similarity relationships in the ALL condition do not fall into clear clusters in two dimensions.
(TIF)

**S2 Fig. Pairwise cosine similarities between object classes in the penultimate CNN layer for each pretraining condition.** Average cosine similarities were calculated from 50 random examples per class. Object attributes are structured periodically in the data set. For object class $c$, color is determined by $((c − 1) \bmod 16)//4$, and shape by $c − 1 \bmod 4$, where mod is the modulo operator, and // division without remainder. These periodic patterns are reflected in the similarity matrices. However, the patterns are not perfect as similarities are still influenced by the input topology and not entirely determined by the label distribution.
(TIF)

**S3 Fig. Performance on the language learning and language emergence task, when language and vision modules are trained.** Shown are boxplots of training and test rewards in the language learning and language emergence scenarios, when studying the influence of differences in language on perception. The plots are generated from the results across ten runs each for communication partners with different perceptual biases (color-coded), always in combination with a DEFAULT agent. In the language learning scenario, the sender (vision and language module) is fixed and we study the effects on the DEFAULT receiver, that is learning the language. In the language emergence scenario, we consider the two cases that a DEFAULT receiver is paired with different senders, and that a DEFAULT sender is paired with different receivers.
(TIF)

**S4 Fig. Control experiments varying task-relevant attributes.** In our control experiments for the evolutionary analysis, we vary which attributes are relevant to the communication game. Always two of the attributes color, scale, and shape are relevant, i.e. one attribute is not relevant. For the irrelevant attribute, sender and receiver target may have different values. Shown are example inputs for different relevance conditions: color irrelevant (top row), scale irrelevant (middle row), and shape irrelevant (bottom row). The receiver target for each condition is marked by a black box.
(TIF)

**S5 Fig. Linguistic biases for the mixed-bias control simulations.** Shown are the effectiveness scores per attribute when combining a sender and a receiver with the same mixed bias. The agents' bias is given on the x-axis, the score on the y-axis, and the attribute for which the score is calculated is indicated by the bar labels. Bars of enforced attributes are dark gray. Results are shown for the three different relevance conditions: (A) color irrelevant, (B) scale irrelevant,

(C) shape irrelevant. We report means and bootstrapped 95% CIs of twenty runs each. Again, the differences in visual perception systematically influence the emerging language. The scores further show that only visual biases for task-relevant attributes are reflected in the language. (TIF)

**S1 Table. Performance of biased-default agent combinations when only the language modules are trained.**
(PDF)

**S1 Appendix. Entropy analysis between target objects, messages, and selections.**
(PDF)

**S2 Appendix. Increasing vocabulary size and number of distractors.**
(PDF)

**S3 Appendix. Control simulations without classification loss.**
(PDF)

**S4 Appendix. Grid search for mixed-bias agents.**
(PDF)

**S5 Appendix. Extension to two senders and two receivers.**
(PDF)

**S6 Appendix. Extension to flexible-role agents.**
(PDF)

## Author Contributions

**Conceptualization:** Xenia Ohmer, Michael Marino, Michael Franke, Peter König.

**Data curation:** Xenia Ohmer, Michael Marino.

**Formal analysis:** Xenia Ohmer.

**Funding acquisition:** Michael Franke, Peter König.

**Methodology:** Xenia Ohmer, Michael Marino.

**Software:** Xenia Ohmer, Michael Marino.

**Supervision:** Michael Franke, Peter König.

**Validation:** Xenia Ohmer, Michael Marino.

**Visualization:** Xenia Ohmer.

**Writing – original draft:** Xenia Ohmer.

**Writing – review & editing:** Xenia Ohmer, Michael Marino, Michael Franke, Peter König.

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
