## [Decision Letter · Decision Letter 0]

14 Jun 2022

Dear Ms Ohmer,

Thank you very much for submitting your manuscript "Mutual influence between language and perception in multi-agent communication games" for consideration at PLOS Computational Biology.

As with all papers reviewed by the journal, your manuscript was reviewed by members of the editorial board and by several independent reviewers. In light of the reviews (below this email), we would like to invite the resubmission of a significantly-revised version that takes into account the reviewers' comments.

The majority of the reviewers appreciated the work. However, there are still major concerns. Among them, reviewer 1's comments reflected a concern that the link to the "big" question of the interaction between language and vision is less strong. Please at least address these comments in the discussion, especially the point about the evolutionary time-scales. Reviewer 3's point that classification cost can introduce bias also appears important: since classification labels unavoidably rely on language, the result does not seem to reveal how communication task directly impact the perceptual system. Can you provide the result when this classification loss is not introduced to give readers a comprehensive understanding of the findings? Although we only highlighted these, please respond to all comments in the next revision.

We cannot make any decision about publication until we have seen the revised manuscript and your response to the reviewers' comments. Your revised manuscript is also likely to be sent to reviewers for further evaluation.

Sincerely,

Ming Bo Cai

Associate Editor

PLOS Computational Biology

Natalia Komarova

Deputy Editor

PLOS Computational Biology

The majority of the reviewers appreciated the work. However, there are still major concerns. Among them, reviewer 1's comments reflected a concern that the link to the "big" question of the interaction between language and vision is less strong. Please at least address these comments in the discussion, especially the point about the evolutionary time-scales. Reviewer 3's point that classification cost can introduce bias also appears important: since classification labels unavoidably rely on language, the result does not seem to reveal how communication task directly impact the perceptual system. Can you provide the result when this classification loss is not introduced to give readers a comprehensive understanding of the findings? Although we only highlighted these, please try to respond to all comments in the next revision. Please feel free to let us know if you have any questions.

Reviewer's Responses to Questions

**Comments to the Authors:**

Reviewer #1: This paper addresses an interesting issue: the potential interactions between visual perception and language. And uses computational simulations in which agents are possess deep convolution neural networks, with the aim of building a model with theoretical implications for the potential mutual interdependence of language and perception. The simulations are well done, and the paper generally clearly written. My concern, though, is that the simulations and their results are not really sufficient to significantly address these broader issues in cognitive science at which they are aimed.

As far as I can tell, the set of stimuli, and categories, is extremely simple (stimuli appear to vary on a small number of dimensions); and the computational problem of categorising these stimuli is close to trivial (i.e., the full power of deep learning is not required). For many purposes, such simplification absolutely fine; but when part of the objective is to understand the potential implications of communication on perception, I suspect it is not.

One reason is as follows: given the incredibly difficult challenge of processing natural images, the visual system is under enormous pressure to organise itself in a way that captures the complex hierarchical structure of those images. And then adding the additional selectional pressure to communicate specific messages could, potentially, pale into insignificance.

This problem is amplified by the fact that the visual brain is strongly constrained by the requirement to control the motor system, and to engage in complex interactions with the world (to know what is a potential predator, or what is edible), irrespective of communication. Again, the impact of the need to communicate may be dwarfed by the intense and continual pressure for the perceptual system to create representations interface naturally with action.

So the theoretical question of interest, when considering the human case in which language may provide additional constraints, has to be considered in this light. Could it be that the pressure of needing to communicate can be powerful enough to partially reshape the vision-internal constraints (and overcome action-relevant constraints)?

Demonstrating that there can, in principle, be an interaction between communication and categorisation just seems to low a bar - I think most computational modelers would assume that this is bound be possible in principle. The real question whether it happens in practice, given the actual informational and communicative constraints imposed by real vision and real language. Using an approach similar to that outlined in the paper, it would surely be possible to explore this kind of question with deep neural networks, given that they are able to do reasonably good categorisation of real visual images, as well as to generate reasonably good natural language outputs. I think this would be much more interesting, and potentially a good contribution to the journal. Of course, drawing conclusions about the human case would still face the challenge that any such model may rather underestimate the degree to which the visual system requires help us navigate the everyday world, as well as communicate with each other (indeed, for a nonhuman animal, this is pretty much all the visual system has to do). But this would be the kind of issue that could be dealt with in discussion, perhaps.

A final more specific theoretical point is that I think the third theoretical goal of the paper, concerning the possible coevolution of human vision and language needs very careful thought. The visual system is very old; and language is both very new by comparison, but also extremely rapidly changing, and, for that matter, the categories used across languages are themselves very variable (just think of the extraordinary variability of colour categories, for example).

So any story in which there is substantial coevolution of the visual system to linguistic categories is going to be very hard to get off the ground--- it would have to assume that there are certain categories that are extremely stable across languages and across time, and old enough to have a substantial influence on genes underlying the neural organization of vision. And this certainly will not be true of many of the categories that we use in everyday life, so many of which refer to an artificial world of objects and institutions that didn’t even exist until the last few centuries (think of chairs, pens, pizzas, offices, etc).

This would seem to imply that language is unlikely to play a major role in shaping vision through biological evolution. It is much more credible that the need to learn the categories of a specific language might, within the lifetime of the individual, influence the structure of the visual system at least to some extent. After all, this is clearly true of learning to read--- the specific neural machinery for recognising letters is biologically “wired in” but obviously not through processes of biological evolution (as literacy is so recent; and the ‘machinery’ is specific to particular scripts/languages).

Overall, this is well conducted work, and a well written manuscript; but it’s overall theoretical contribution seems not to be sufficient for publication in PLOS Computational Biology.

Reviewer #2: I was very impressed by Ohmer et al’s manuscript. It demonstrates excellent scholarship — bringing in much of the key literature. For example, I made a note on p. 4 indicating the high relevance of Cangelosi et al’s. now fairly old work on “sensorimotor toil” vs “symbolic theft” and was impressed to see it cited later on! I’ve done some conceptually similar modeling as part of a larger “language augmented cognition” project, e.g.,

Lupyan, G. (2012). What do words do? Towards a theory of language-augmented thought. In B. H. Ross (Ed.), The Psychology of Learning and Motivation (Vol. 57, pp. 255–297). Academic Press. http://www.sciencedirect.com/science/article/pii/B9780123942937000078

( http://sapir.psych.wisc.edu/papers/lupyan_2012_languageAugmentedThought.pdf )

but the authors go way beyond the toy models presented in that work.

The manuscript is also very clear and thorough in its presentation of methods and results. This is an impressive project especially considering the student first authors. Below are some relatively minor comments.

1. The many functions of vision

“To study the influence of language on perception, we allow agents to adapt their visual representations (CNN weights) while playing the communication game.” — this is a fine approach, but the authors should be more explicit that vision has many functions; yielding representations that we can (however approximately) talk about, is just one function. Hence, we are highly adept at distinguishing small differences hue even though we cannot easily put them in words. The categorical effects on perception that we do see (especially in vision) are rather slight, e.g., categorical color perception goes away entirely in some tasks, eg.,

Witzel, C., & Gegenfurtner, K. R. (2015). Categorical facilitation with equally discriminable colors. Journal of Vision, 15(8), 22–22. https://doi.org/10.1167/15.8.22

and while language enhances categorical perception ,e.g., as we describe in this paper:

Forder, L., & Lupyan, G. (2019). Hearing words changes color perception: Facilitation of color discrimination by verbal and visual cues. Journal of Experimental Psychology: General, 148(7), 1105–1123. https://doi.org/10.1037/xge0000560

in the paper above, you’ll note that on the no-label trials people’s discrimination is about equal within and between categories.

So, if an agent’s only goal is to communicate about a discrete set of categories, you’d expect perception to just represent things categorically. But because categorization/naming/communication are just one goal of perception, perceptual systems continue to represent “category irrelevant” properties. Because in your model the perception is *for* communication, the effects of having to communicate about this or that dimension likely over-estimate the kinds of effects of language on perception one tends to observe.

2. Enhanced perception?

“Enhanced perception of relevant features is evolutionary stable. “ (p. 23).

I would have liked to see some discussion of *what* the authors think is being enhanced. If the models here are taken as very roughly representing human visual systems, what about perception is being enhanced? And how “perceptual” is it? For example, “enhanced perception” can be achieved purely through attentional modulation. Although attention largely works by altering perceptual representations through the processing hierarchy, there is a difference between “better” perception due to a high level attentional selection mechanism vs. a dynamic retuning of receptive fields of primary sensory neurons. If you think the influence of communication/lanbguage on perception is not attentional (the current models after all lack attentional mechanisms), are you thinking of a more permanent “warping” of low-level visual representations? Or are the differences more likely to be observed higher up in the processing hierarchy? If the latter, are these changes still “perceptual” (rather than more decisional)? (Many people care about these differences, e.g., as they relate to questions about cognitive penetrability of (especially low-level) perception).

3. Some additional relevant literature

If you are not familiar with it, you may be interested in the following papers:

Ozgen, E., & Davies, I. R. L. (2002). Acquisition of categorical color perception: A perceptual learning approach to the linguistic relativity hypothesis. *Journal of Experimental Psychology-General*, *131*(4), 477–493.

You cite some work by Luc Steels, but this key paper is an omission:

Steels, L., & Belpaeme, T. (2005). Coordinating Perceptually Grounded Categories Through Language: A Case Study for Colour. Behavioral and Brain Sciences, 28(04), 469–489. https://doi.org/10.1017/S0140525X05000087

Finally, you may be interested in this recent paper where we investigated the effects of different types of labels on DNN visual representations and fit to human behavior:

Ahn, S., Zelinsky, G. J., & Lupyan, G. (2021). Use of superordinate labels yields more robust and human-like visual representations in convolutional neural networks. Journal of Vision, 21(13), 13. https://doi.org/10.1167/jov.21.13.13

With regard to this point, “supervised learning of symbolic object descriptions warped these similarity spaces even further, leading to increasingly categorical perception”, you may be interested in this empirical paper:

Lupyan, G. (2008). The conceptual grouping effect: Categories matter (and named categories matter more). *Cognition*, *108*(2), 566–577.

Other comments:

a. “Our simulations mirror several empirically observed phenomena: labels group together visually similar objects, “ — it may be worth clarifying what kinds of labels you are thinking of. This is true for labels like ”circle” and “bowl” and “striped”. It’s not so true for arguably the bulk of language. For example, none of the words I am using in this sentence really have anything to do with perceptual properties. And much of the language we use — especially among adults — has this property. We talk about social relationships, plans, desires, things removed in time — the categories denoted by these words can’t really be described in terms of perceptual features (at least not in any straightforward way).

-Gary Lupyan

Reviewer #3: The authors explore the influences of communicative interactions on perception and the influences of perceptual biases on the emergent communication protocols. The authors use communication game simulations with two agents. They find that agents’ perception adapts to the communication-relevant attributes, that emergent communication is biased towards perceptually enhanced dimensions, and that possessing a communication-relevant perceptual bias presents an evolutionary advantage.

Ohmer and colleagues’ approach is novel, and it introduces an exciting new direction of research on both 1) language evolution and 2) effects of language on perception. So far, the area of language influences on perception has focused on one-agent simulations and experiments. Simulations of language evolution, on the other hand, have mainly used the agents that have no access to perceptual properties of the objects, or the agents’ whose perception is not adaptive. This study is methodologically rigorous and extremely well done. The authors’ main conclusions are overall well substantiated by their experiments. The paper is well written. This paper presents a complete investigation of the target questions, so I only have suggestions regarding authors’ presentation of the approach and interpretation of the results. I am happy to recommend this paper for publication in PLOS Computational Biology.

MAJOR COMMENTS

1. Generalizability. The authors investigate two-way influences of perceptual biases and communication by using two-agent simulations whose communicative roles (speaker and listener) are fixed. This limitation in the approach does not undermine the originality and rigor of this work, but it may limit authors’ conclusions: for example, it has been shown that many results from two-agent and fixed-role simulations are not guaranteed to generalize to “more than 2”-agent cases. I recommend the authors to reflect on the generalizability of their results to “more than two”- and “flexible role”-agents (each agent learns to both speak and listen) scenarios in the discussion section.

2. Classification confounds. On page 11 (lines 201-205), the authors describe adding classification loss to the agents’ training to avoid their perceptual systems from “deteriorating” when fully adapting to the communication task. I would appreciate it if the authors clarify why such “hyper-adaptation” presents a problem for their experiments, as their target question is perceptual adaptation in response to communicative pressures, at least in experiment 2. More importantly, I expect that adding the classification task does not only attenuate the effects of communication on perception (as the authors note), but it also creates perceptual effects on its own (e.g. potential bias for representing color rather than shape or scale). I recommend the authors to explicitly note this for all the effects that can be potentially attributable to the classification task in the experiments 1 and 2 (I see that this confound has been explicitly addressed for experiment 3). For example, on page 20 (lines 322-323), the authors describe one (potential) effect of the classification subtask:

“In the default condition (left) the messages are strongly grounded in object color, which can be attributed to the inherent color bias of the default CNN” (this alternative, classification task-based, interpretation might also require authors to correct the discussion in lines 557-560)

MINOR COMMENTS

When introducing or discussing the effects of perceptual biases on communication, I recommend the authors to refer to Morten Christiansen’s work to better frame their contribution within the current hypotheses on the drivers of language evolution:

Christiansen, M. H., & Chater, N. (2008). Language as shaped by the brain. Behavioral and brain sciences, 31(5), 489-509.

Lines 22-24, page 3: “The formation of linguistic expressions is strongly influenced by perception, not only for concrete concepts like colors [17] but also abstract ones [18]”. This sentence is confusing. Do the authors mean perception for concrete “attributes” (=color)? How does “perception” of abstract “concepts” look like? I recommend rewriting or clarifying this sentence.

Line 76, page 5: authors use the term “natural selection”: I am not sure if it is appropriate in this context. Authors’ evolutionary simulations indeed reveal the “selection” pressures, but such pressures can be realized in either “natural” or “cultural” evolution.

Caption to Fig.1, page 8: “Simultaneously, the receiver’s vision module processes the images of the target and the distractor images.” The use of “simultaneously” is confusing here: do the authors mean that all the images were processed at the same time? I assume this is not the intended interpretation as the receiver’s vision module seems to only have one processing stream.

Fig. 2, page 10: This figure might be very helpful for understanding relational smoothing, but it was difficult for me to understand it. I recommend the authors to revise it or its caption to make it more accessible.

Fig. 6, page 21: The bars and labels on this figure are confusing. Do the x-axes represent sender/receiver CNN biases’ conditions?

Lines 642-643, page 30: this sentence is not clear. What kind of “interaction” do the authors have in mind?

Best,

Marina Dubova

**Have the authors made all data and (if applicable) computational code underlying the findings in their manuscript fully available?**

Reviewer #1: None

Reviewer #2: Yes

Reviewer #3: Yes

PLOS authors have the option to publish the peer review history of their article (what does this mean?). If published, this will include your full peer review and any attached files.

Reviewer #1: No

Reviewer #2: **Yes: **Gary Lupyan

Reviewer #3: **Yes: **Marina Dubova
---

## [Decision Letter · Decision Letter 1]

30 Aug 2022

Dear Ms Ohmer,

Thank you very much for submitting your manuscript "Mutual influence between language and perception in multi-agent communication games" for consideration at PLOS Computational Biology. As with all papers reviewed by the journal, your manuscript was reviewed by members of the editorial board and by several independent reviewers. The reviewers appreciated the attention to an important topic. Based on the reviews, we are likely to accept this manuscript for publication, providing that you modify the manuscript according to the review recommendations.

Sincerely,

Ming Bo Cai

Academic Editor

PLOS Computational Biology

Natalia Komarova

Section Editor

PLOS Computational Biology

[LINK]

Reviewer's Responses to Questions

**Comments to the Authors:**

Reviewer #1: This is high quality work. My main concern on the first round was that the contribution is not quite substantial enough to be useful - because the basic pattern of results feels more or less inevitable (if you impose a communicative task, a 'visual system' will be shaped by it); the real question, to me, is the relative informational demands of vision-for-action and vision-for-communication in natural language.

My concern remains - but in the light of the feeling from the other reviewers that this is a sufficient contribution, and perhaps a stimulus to further work, I am happy to recommend acceptance. I do hope the authors (or another team) pick up this line of work using more realistic images, to address the informational question more thoroughly in the future.

I appreciate the adjustment to the paper regarding the evolutionary picture.

Reviewer #2: #review

PCOMPBIOL-D-22-00198R1

I commend the authors for such a thorough revision. I only have a few remaining comments:

1. “ language manipulates judgments of perceptual similarity by imposing categorical structure [20]” — the better word choice here would be “affects” in place of “manipulates”. More importantly though, the cited work is quite old (and flawed) and has been superseded by better controlled studies that get at effects of language on visual processing rather than higher-level judgments e.g.:

Winawer, J., Witthoft, N., Frank, M. C., Wu, L., Wade, A. R., & Boroditsky, L. (2007). Russian blues reveal effects of language on color discrimination. *Proceedings of the National Academy of Sciences of the United States of America*, *104*(19), 7780–7785.

Winawer, J., Witthoft, N., Frank, M. C., Wu, L., Wade, A. R., & Boroditsky, L. (2007). Russian blues reveal effects of language on color discrimination. *Proceedings of the National Academy of Sciences of the United States of America*, *104*(19), 7780–7785.

Forder, L., & Lupyan, G. (2019). Hearing words changes color perception: Facilitation of color discrimination by verbal and visual cues. Journal of Experimental Psychology: General, 148(7), 1105–1123. https://doi.org/10.1037/xge0000560

2. I would like the authors to further clarify how they think of classification loss. Here is where I am coming from. In the case of naming an object, the category comes from language. Where are the categories that are being used in the non-communicative classification task coming from (conceptually)? I.e., when you writ “ In the case of these “suboptimal” solutions, the classification loss guarantees that the visual representations still contain information that is sufficient to classify each object” — classify as what, exactly?

-Gary Lupyan

Reviewer #3: The revised version of the manuscript has further improved an already excellent work. All the reviewers' comments seem to be addressed, either completely or in essence. My own concerns are fully resolved -- I especially appreciate authors' efforts to further explore the generalizability of their results with additional simulations. I am looking forward to recommending this work to my colleagues as a reference, once it is published.

**Have the authors made all data and (if applicable) computational code underlying the findings in their manuscript fully available?**

Reviewer #1: None

Reviewer #2: Yes

Reviewer #3: Yes

PLOS authors have the option to publish the peer review history of their article (what does this mean?). If published, this will include your full peer review and any attached files.

Reviewer #1: No

Reviewer #2: No

Reviewer #3: **Yes: **Marina Dubova

Figure Files:

Data Requirements:

Reproducibility:

References:

---

## [Decision Letter · Decision Letter 2]

14 Oct 2022

Dear Ms Ohmer,

We are pleased to inform you that your manuscript 'Mutual influence between language and perception in multi-agent communication games' has been provisionally accepted for publication in PLOS Computational Biology.

Best regards,

Ming Bo Cai

Academic Editor

PLOS Computational Biology

Natalia Komarova

Section Editor

PLOS Computational Biology

Thanks for the effort of revision.

Reviewer's Responses to Questions

**Comments to the Authors:**

Reviewer #2: Great job with the revision!

**Have the authors made all data and (if applicable) computational code underlying the findings in their manuscript fully available?**

Reviewer #2: Yes

PLOS authors have the option to publish the peer review history of their article (what does this mean?). If published, this will include your full peer review and any attached files.

Reviewer #2: **Yes: **Gary Lupyan

---

## [Editor Report · Acceptance letter]

24 Oct 2022

PCOMPBIOL-D-22-00198R2 

Mutual influence between language and perception in multi-agent communication games

Dear Dr Ohmer,

I am pleased to inform you that your manuscript has been formally accepted for publication in PLOS Computational Biology. Your manuscript is now with our production department and you will be notified of the publication date in due course.

With kind regards,

Zsofi Zombor
